# A Novel Method Based on Multi-Island Genetic Algorithm Improved Variational Mode Decomposition and Multi-Features for Fault Diagnosis of Rolling Bearing

**DOI:** 10.3390/e22090995

**Published:** 2020-09-07

**Authors:** Tao Liang, Hao Lu

**Affiliations:** School of Artificial Intelligence and Data Science, Hebei University of Technology, Tianjin 300401, China; 18332901650@163.com

**Keywords:** rolling bearing, fault diagnosis, variational mode decomposition, multi-island genetic algorithm, multi-features

## Abstract

Aiming at the problem that it is difficult to extract fault features from the nonlinear and non-stationary vibration signals of wind turbine rolling bearings, which leads to the low diagnosis and recognition rate, a feature extraction method based on multi-island genetic algorithm (MIGA) improved variational mode decomposition (VMD) and multi-features is proposed. The decomposition effect of the VMD method is limited by the number of decompositions and the selection of penalty factors. This paper uses MIGA to optimize the parameters. The improved VMD method is used to decompose the vibration signal into a number of intrinsic mode functions (IMF), and a group of components containing the most information is selected through the Holder coefficient. For these components, multi-features based on Renyi entropy feature, singular value feature, and Hjorth parameter feature are extracted as the final feature vector, which is input to the classifier to realize the fault diagnosis of rolling bearing. The experimental results prove that the proposed method can more effectively extract the fault characteristics of rolling bearings. The fault diagnosis model based on this method can accurately identify bearing signals of 16 different fault types, severity, and damage points.

## 1. Introduction

The failure of the rolling bearing of the wind turbine is one of the important factors affecting the normal operation of the wind turbine, so it is very important to diagnose the running state of the rolling bearing in time in actual production [1]. When the bearing fails, the vibration signal characteristics will also change accordingly. If the signal characteristics of each state can be extracted, the bearing fault diagnosis can be carried out.

Empirical mode decomposition (EMD) has been widely used in mechanical fault diagnosis as an adaptive signal decomposition method [2]. Wang et al. [3] use empirical mode decomposition to decompose the vibration signal into a finite number of eigenmode functions and a residual function, extracts the main eigenmode function energy and its local average frequency characteristics, and finally uses the composite eigenvector as the input of the principal component analysis classifier completes the identification of the fault. Zhou et al. [4] combined EMD and singular value decomposition (SVD) to extract bearing fault feature vectors, and used fuzzy c-means (FCM) clustering to identify rolling bearing fault types. Through the above methods, the importance of EMD in bearing fault diagnosis is proved. Although EMD can extract the features of mechanical vibration signals, the extracted signal features are not accurate enough due to the shortcomings of the EMD method such as modal aliasing. In order to overcome these shortcomings of EMD, Wu et al. [5] presented a new ensemble empirical mode decomposition (EEMD) method, applying a noise-assisted data analysis method to improve the original EMD. Li et al. [6] used the local mean decomposition (LMD) method to decompose the bearing vibration signal and then performed subsequent bearing fault diagnosis. In addition, there are other methods, such as CEEMD, etc., but the above-mentioned improved method only suppresses modal aliasing and is far from completely eliminated.

In 2014, Dragomiretskiy et al. [7] proposed a new adaptive signal processing method, VMD. As a new adaptive signal processing method, VMD method introduces the variational model to transform the signal decomposition into the optimization problem of the optimal solution of the constraint model, which can avoid the end effect, restrain the mode confusion, and has high decomposition efficiency [8]. Liu et al. [9] proposed a feature extraction method based on VMD and SVD, using standard FCM clustering for fault identification. Ding et al. [10] proposed a rolling bearing fault diagnosis method combining VMD decomposition and t-SNE manifold learning. It uses the VMD method to decompose the original vibration signal of the rolling bearing into several IMFs, then calculates the time-frequency characteristics of each modal component to form high-dimensional fault features and uses t-SNE to extract secondary features of the fault to obtain low-dimensional sensitive features. Finally, the low-dimensional sensitive features are used as the input of the K-means classifier to realize the identification of fault types. However, in the above studies, when using the VMD method to decompose bearing vibration signals, the selection of the number of decompositions and the penalty parameters are based on experience rather than a reasonable basis. In order to solve this problem reasonably, this paper takes the local minimum envelope entropy of the modal component as the objective function and uses MIGA algorithm to search for the optimal parameter combination of VMD. On this basis, a VMD method based on MIGA algorithm is proposed, which can extract accurate fault characteristic information from fault signals.

With the development of artificial intelligence, bearing fault diagnosis is increasingly treated as a category of pattern recognition. Its effectiveness and reliability largely depend on the choice of feature vector. Dou et al. [11] decomposed the bearing vibration signal into a set of IMFs through EEMD, and calculated the Renyi entropy of each IMF component as the fault feature vector. Li et al. [12] use the EEMD method to adaptively decompose the bearing vibration signal into multiple eigenmode function components. Then six eigenmode function components are selected according to the kurtosis criterion, and the matrix formed by them is subjected to singular value decomposition to obtain the fault feature vector. Zhou et al. [13] performed EMD decomposition on the original vibration signal to obtain representative IMFs, and calculated the Hjorth parameter as the fault feature vector. Therefore, this paper takes the multi-features based on Renyi entropy feature, singular value feature, and Hjorth parameter feature as the fault feature vector of rolling bearing vibration signal.

Based on this, this paper takes rolling bearing vibration signals as the research object, and innovatively proposes a feature extraction method based on MIGA-VMD decomposition and multi-features. The experimental results show that this method can effectively improve the accuracy of fault diagnosis of rolling bearings.

The remainder of this paper is organized as follows: Section 2 introduces the decomposition principle of the VMD method. Section 3 uses the multi-island genetic algorithm to optimize parameters and obtain the optimal parameter combination. Section 4 provides a brief introduction to the Holder coefficient. Section 5 provides an overview of multi-features. Section 6 elaborates the fault diagnosis steps of rolling bearing based on MIGA-VMD decomposition and multi-features and conducts simulation experiments in Section 7. Section 8 provides the conclusion of the article.

## 2. Brief Introduction of VMD

The process of variational modal decomposition is essentially the process of constructing variational problems and solving variational problems [9]. This section introduces the VMD method from the two aspects of construction and decomposition.

### 2.1. Construction of the Variational Problem

The VMD algorithm redefines the decomposed IMF component as an AM-FM signal. Assuming that the original signal is decomposed into K IMF components, the expression of the kth IMF component is
(1)$$u_{k}\left(t\right)=A_{k}\left(t\right)\cos\left[\phi_{k}\left(t\right)\right],$$
where k is the integer between 1∼K, $\phi_{k}\left(t\right)$ is a non-decreasing phase function; $A_{k}\left(t\right)$ represents the envelope function.

First, for each IMF component, use the Hilbert transform to find its analytic signal to obtain a unilateral spectrum, and the spectrum expression is
(2)$$\left[\delta \left(t\right)+\frac{j}{\pi t}\right]*u_{k}\left(t\right)$$

Secondly, according to the estimated center frequency of the hybrid, modulate the spectrum to the corresponding base frequency band, which is recorded as
(3)$$\left[\left(\delta \left(t\right)+\frac{j}{\pi t}\right)*u_{k}\left(t\right)\right]*e^{-j\omega_{k}t}$$

Finally, calculate the square of the $L_{2}$ norm of the time gradient of the demodulated signal, estimate the bandwidth of the modal component, and introduce constraints to construct a constrained variational model, which can be expressed as
(4)$$\left\{ \begin{array}{l} \underset{\left\{u_{k}\right\},\left\{\omega_{k}\right\}}{\min}\left\{\sum_{k=1}^{K}{\left\|\partial \left(t\right)\left[\left(\delta \left(t\right)+\frac{j}{\pi t}\right)*u_{k}\left(t\right)\right]*e^{-j\omega_{k}t}\right\|}_{2}^{2}\right\}\\ s.t.\sum_{k=1}^{K}u_{k}=x \end{array}\right.,$$
where $\left\{u_{k}\right\}$ represents the K IMF components obtained by VMD decomposition; $\left\{\omega_{k}\right\}$ represents the center frequency corresponding to the IMF component; δ(t) is the Dirichlet function; * is the convolution operation; x represents the original signal.

### 2.2. Solving the Variational Problem

The penalty parameter α and the Lagrangian multiplication coefficient λ(t) are introduced to change the constrained variational problem into an unconstrained variational problem. Among them, the secondary penalty factor can ensure the accuracy of signal reconstruction in the presence of Gaussian noise, and the Lagrangian coefficient keeps the constraint conditions strict. The augmented Lagrangian function is
(5)$$\begin{array}{l} L\left(\left\{u_{k}\right\},\left\{\omega_{k}\right\},\lambda \right)=\\ \langle \lambda \left(t\right),x\left(t\right)-\sum_{k}u_{k}\left(t\right)\rangle +\\ {\left\|x\left(t\right)-\sum_{k}u_{k}\left(t\right)\right\|}_{2}^{2}+\alpha \sum_{k}{\left\|\partial_{t}\left(\sigma \left(t\right)+\frac{j}{\pi t}\right)u_{k}\left(t\right)e^{-j\omega_{k}t}\right\|}_{2}^{2} \end{array}$$

Then use the alternating direction multiplier method (ADMM) to solve the extremum of the above equation to obtain the frequency domain expression of the IMF component as
(6)$${\hat{u}}_{k}^{n+1}\left(\omega \right)=\frac{\hat{x}\left(\omega \right)-\sum_{i=1,i\neq k}^{K}{\hat{u}}_{i}\left(\omega \right)+\frac{\hat{\lambda }\left(\omega \right)}{2}}{1+2\alpha {\left(\omega -\omega_{k}\right)}^{2}},$$
where ${\hat{u}}_{k}^{n+1}\left(\omega \right)$ represents the Wiener filter of the current remaining amount $\hat{x}\left(\omega \right)-\overset{k}{\underset{i=1,i\neq k}{\sum }}{\hat{u}}_{i\left(\omega \right)}$. Inverse Fourier transform is performed on the Wiener filter, and the real part obtained is the time domain signal $u_{k}\left(t\right)$. Converting the center frequency problem to the frequency domain, the update formula of the center frequency $\omega_{k}^{n+1}$ is obtained as
(7)$$\omega_{k}^{n+1}=\frac{\int_{0}^{\infty }\omega {\left|{\hat{u}}_{k}\left(\omega \right)\right|}^{2}d\omega }{\int_{0}^{\infty }{\left|{\hat{u}}_{k}\left(\omega \right)\right|}^{2}d\omega },$$
where $\omega_{k}^{n+1}$ represents the center of gravity of the current IMF component power spectrum. The update formula of $\lambda^{n+1}$ is
(8)$$\lambda^{n+1}=\lambda^{n}+\tau \left(x-\sum_{k}u_{k}^{n+1}\right)$$

Through the above analysis, the algorithm flow of VMD is as follows:

Step 1: Initialize $\left\{u_{k}^{1}\right\}$, $\left\{\omega_{k}^{1}\right\}$, $\lambda_{k}^{1}$, and n;

Step 2: Update $u_{k}$ and $\omega_{k}$ according to Equations (6) and (7);

Step 3: Update λ according to Equation (8);

Step 4: Set a judgment accuracy e>0, if $\underset{k}{\sum }\|{\hat{u}}_{k}^{n+1}-{\hat{u}}_{k}^{n}\|{}_{2}^{2}/\|{\hat{u}}_{k}^{n}\|{}_{2}^{2}<e$, stop the iteration, otherwise return to step 2.

## 3. Parameter Optimization Based on Multi-Island Genetic Algorithm

In the VMD decomposition process, the number of decomposed IMF components K and the penalty parameter α have a greater impact on the decomposition result. If the K value is too large, false components will be generated in the decomposition results, which will mislead the judgment of the experimental results; if the K value is small, modal aliasing will occur. If the penalty parameter α is smaller, the bandwidth of each IMF component after decomposition is smaller. On the contrary, if the penalty parameter α is larger, the bandwidth of each IMF component after decomposition is larger. Except for K and α, other parameters have little influence on the decomposition effect and are set as empirical values, namely tau=0, init=1, DC=0, ε=1e−7. Therefore, before performing VMD decomposition on the signal, appropriate K and α need to be selected.

In the process of optimizing VMD parameters, particle swarm optimization (PSO) algorithm and frog jumping algorithm are used to optimize VMD parameters [14]. However, the PSO algorithm is easy to fall into the local optimum and the search path is more complicated. The convergence speed of the frog jumping algorithm is slow, and it may fall into a local optimal value. These two algorithms can only determine the number of modal components and the calculation process is also more complicated [15]. The concept of genetic algorithm was first put forward by Holland of the University of Michigan in 1975, imitating the law of “survival of the fittest” in nature to carry out the optimization process [13]. In the parameter optimization process, the genetic algorithm is used to optimize the parameters of VMD. It obtains parameter optimization solutions through iterative operations such as chromosome selection, crossover, mutation, etc. This algorithm has the characteristics of implicit parallel search and has a strong global search ability in the process of solving, but the genetic algorithm is easy to converge prematurely and fall into local optimum. To solve the problem of premature convergence and local optimal solution that often occurs with genetic algorithm, some scholars have proposed a multi-island genetic algorithm. The multi-island genetic algorithm uses the operators and selection principles of the genetic algorithm repeatedly and appropriately, from parents to offspring, from offspring to grandchildren, continue to reproduce, so that the adaptability of the population to the environment continues to improve. In this paper, multi-island genetic algorithm is used for parameter optimization to obtain the optimal parameter combination.

Table 1 lists the specific steps of the multi-island genetic algorithm.

The parameter design of the multi-island genetic algorithm is as follows: the initial population is divided into 10 islands, the population number of each island is set to 10, and the number of optimized generations is set to 10, so the calculated number of individuals is 1000. The mutation probability is set to 0.01, the crossover probability is set to 1, the migration probability is set to 0.01, and the number of elite individuals is set to 1.

In order to verify the feasibility of multi-island genetic algorithm, it is compared with genetic algorithm. According to the characteristics of the fault vibration signal of rolling bearing, a simulation signal y(u) is constructed. The expression of the simulated signal is given in Formula (9).
(9)$$y\left(u\right)=-5sinu_{1}sinu_{2}sinu_{3}sinu_{4}sinu_{5}-sin5u_{1}sin5u_{2}sin5u_{3}sin5u_{4}sin5u_{5}+8$$

In this simulation experiment, expression (9) is regarded as the fitness function of genetic algorithm and multi-island genetic algorithm. For genetic algorithms, the number of iterations is 30, the population size is 100, the crossover probability is 1, and the mutation probability is 0.01. For the multi-island genetic algorithm, the initial population is divided into 10 islands, the population number of each island is set to 10, the number of iterations is 30, the crossover probability is 1, the mutation probability is 0.01, the migration probability is 0.01, and the number of elite individuals is 1. The iterative curve of the optimization process is shown in Figure 1 and Figure 2.

From the results of Figure 1 and Figure 2, it can be seen that the genetic algorithm is stuck in premature convergence and the convergence speed is gradually reduced. After the completion of 30 iterations, the obtained result still has a large deviation from the optimal value. Under the same population size, the multi-island genetic algorithm obtains the approximate optimal solution in the 10th iteration. Its convergence speed is faster, the obtained optimal value is closer to the theoretical optimal value, and the accuracy of parameter optimization is better than that of traditional genetic algorithm. The result shows that the multi-island genetic algorithm has a faster convergence speed and stronger global optimization ability.

When using multi-island genetic algorithm for parameter optimization, a fitness function needs to be selected. Reference [16] proposed the concept of envelope entropy $E_{p}$. After the fault signal is decomposed by VMD, the envelope entropy $E_{p}$ of the IMF component can reflect the sparsity of the component. If the IMF component contains more noise after decomposition, thereby concealing the impact characteristics of the fault, the sparsity of the IMF component is weaker and the envelope entropy is larger; on the contrary, if the IMF component contains more regular fault impacts, then the IMF component has strong sparsity, and the envelope entropy is small. Under the influence of the parameter combination K and α, K components will have K envelope entropy values, and the smallest one of the K envelope entropy values is selected as the local minimum envelope entropy $\min E_{p}$. The component corresponding to the minimum entropy value has abundant fault characteristic information [7]. Taking the local minimum entropy as the fitness function in the multi-island genetic algorithm, the entire search process is to find the optimal parameter combination [K,α] corresponding to the global minimum envelope entropy. Therefore, construct the following fitness function
(10)$$\min F=\min E_{p},$$
the expression of envelope entropy $E_{p}$ is
(11)$$\left\{ \begin{array}{l} E_{p}=-\sum_{j=1}^{N}e_{j}\mathrm{lg}e_{j}\\ e_{i}=a\left(j\right)/\sum_{j=1}^{N}a\left(j\right) \end{array}\right.,$$
where j=1,2,3,…,N, a(j) is the envelope signal obtained by Hilbert demodulation of x(j), $e_{j}$ is obtained by normalizing a(j), and $E_{p}$ is obtained according to the calculation rules of information entropy.

In order to verify the effectiveness of the local envelope entropy as a fitness function, a simulation signal of the rolling bearing is constructed. Its expression is
(12)$$c\left(t\right)=y_{0}e^{-2\pi f_{n}\xi t}\sin\left(2\pi f_{n}\sqrt{1-\xi^{2}}t\right)+n\left(t\right),$$
where c(t) represents the noise-added rolling bearing simulation signal, $y_{0}$ is the displacement constant, $f_{n}$ is the natural frequency of the rolling bearing, ξ is the damping coefficient, n(t) is the Gaussian white noise tending to the real noise, t is the sampling time, sampling Frequency $f_{s}=20\mathrm{KHZ}$, number of sampling points N=4096. Here, we set $y_{0}=3$, $f_{n}=3000\mathrm{HZ}$, and ξ=0.09.

Figure 3 shows the waveforms of the simulation signal under the conditions of no noise, noise intensity = 1, noise intensity = 3, and noise intensity = 5, respectively. By comparing the four pictures, we can see that the periodic pulses in the simulation signal are completely covered by strong noise. There is a positive proportional relationship between the intensity of noise and envelope entropy: as the noise becomes stronger, the periodic pulse becomes blurred and the envelope entropy increases; with the weakening of noise, the periodic pulse becomes prominent and the envelope entropy becomes smaller. This also proves the law that the greater the sparsity of the signal, the smaller the envelope entropy. Therefore, the modal component with minimum envelope entropy can be identified as the optimal component.

## 4. Holder Coefficient

The Holder coefficient can be used to analyze the correlation between two discrete signals. The Holder coefficient algorithm evolves from the Holder inequality [17,18]. The definition of the Holder inequality can be described as [19].

For any vector $X={\left[x_{1},x_{2},\cdots ,x_{N}\right]}^{T}$ and $\;Y={\left[y_{1},y_{2},\cdots ,y_{N}\right]}^{T}$, they satisfy:(13)$$\sum_{i=1}^{N}\left|x_{i}\cdot y_{i}\right|\leq {\left(\sum_{i=1}^{N}{\left|x_{i}\right|}^{p}\right)}^{1/p}\cdot {\left(\sum_{i=1}^{N}{\left|y_{i}\right|}^{q}\right)}^{1/q},$$
where 1/p+1/q=1 and p, q>1.

Based on the Holder inequality, for two discrete signal $x\left(t\right)=\left[x_{1},x_{2},\cdots ,x_{N}\right]$ and $y\left(t\right)=\left[y_{1},y_{2},\cdots ,y_{N}\right]$, the Holder coefficient of these two discrete signal is given by
(14)$$H_{c}=\frac{\sum_{i=1}^{N}x_{i}y_{i}}{{\left(\sum_{i=1}^{N}x^{p}\right)}^{1/p}\cdot {\left(\sum_{i=1}^{N}y^{q}\right)}^{1/q}},$$
where 1/p+1/q=1 and p, q>1; $0\leq H_{c}\leq 1$.

The greater the value of Holder coefficient, the greater the correlation between the two discrete signals. In this paper, the partial IMF components which are closely related to the original signal are selected by calculating the Holder coefficients of each IMF component and the original signal.

## 5. Multi-Features

In order to meet the requirements of accurate diagnosis of rolling bearing fault types, a multi-features feature extraction method based on Renyi entropy feature, singular value feature and Hjorth parameter feature is proposed to extract the health state feature vector from the bearing vibration signal.

### 5.1. Renyi Entropy Feature

For a discrete random variable $X=\left\{x_{k}\;|\;k=1,2,\cdots ,n\right\}$, Renyi entropy is defined as
(15)$$R_{\alpha }\left(X\right)=\frac{1}{1-\alpha }\sum_{i=1}^{n}\log p_{k}^{\alpha },$$
where α represents the order of Renyi entropy; $p_{k}$ is the probability density of event $X=x_{k}$.

In information theory, Renyi entropy, like Shannon entropy, is also a method of quantitatively describing signal information, which can reflect the amount of information and complexity of the signal. For the diagnosis of mechanical equipment, the different operating states of the equipment represent different internal mode complexity. The greater the complexity, the greater the Renyi entropy, especially for some specific faults. Because the fault information is often concentrated in a certain sensitive frequency band, when a fault occurs, the vibration signal in the sensitive frequency band will change significantly, that is, the complexity of the frequency band also changes [11]. Because VMD can be adaptively decomposed into the sum of several IMF components located in different frequency bands and representing different intrinsic modes according to the characteristics of the signal itself, the Renyi entropy of each IMF component of the signal under different operating conditions can reflect the complexity of the frequency band mode to a certain extent, which also provides a basis for the extraction of fault information feature vectors.

The Renyi entropy feature $E=\left[R_{1},R_{2},\cdots ,R_{j}\right]\left(j=1,2,\cdots ,n\right)$ is used as part of the feature vector of rolling bearing fault identification.

### 5.2. Singular Value Feature

Singular value decomposition can decompose the matrix containing signal feature information into different subspaces, and is a feature extraction method that can keep signal features relatively stable under disturbance and noise. At the same time, its singular value can reflect the inherent properties of the matrix and its principal component relationship [20].

For any matrix Φ, there are orthogonal (or unitary) matrices U and V, so that equation (16) holds.
(16)$$\Phi =U\Sigma V^{T}\;\mathrm{or}\;\Phi =U\Sigma V^{H},$$
where $\sum^{\;}=\left[\begin{array}{cc} {\sum^{\;}}_{r} & 0\\ 0 & 0 \end{array}\right]$, and ${\sum^{\;}}_{r}=\mathrm{diag}\left(\sigma_{1},\sigma_{2},\cdots ,\sigma_{r}\right)$, the diagonal elements are arranged in descending order $\left(\sigma_{1}\geq \sigma_{2}\geq \cdots \geq \sigma_{r}\geq 0\right)$, r is the rank of matrix Φ, $\sigma_{1},\sigma_{2},\cdots ,\sigma_{r}$ are the singular values of matrix Φ.

VMD decomposition not only performs adaptive frequency band information description on the signal, but also greatly increases the amount of data. Therefore, singular value decomposition can be used to compress the dimensions of high-dimensional related time series, that is, the singular values $\sigma_{1},\sigma_{2},\cdots ,\sigma_{r}$ are the enrichment of each IMF component information [21].

The singular value feature $S=\left[\sigma_{1},\sigma_{2},\cdots ,\sigma_{r}\right]$ is used as part of the feature vector of rolling bearing fault identification.

### 5.3. Hjorth Parameters Feature

The three parameters proposed by Hjorth are activity, mobility, and complexity [22].

#### 5.3.1. Activity Parameter

The activity parameter represents the signal power and it is defined as the variance of the vibration signal amplitude. If $x\left(t\right)=\left[x_{1},x_{2},\cdots ,x_{N}\right]$ is the sampled vibration signal, then the activity parameter of x(t) is given by
(17)$$Activity\left(x\left(t\right)\right)=variance\left(x\left(t\right)\right)=\sigma_{x}^{2},$$
where $\sigma_{x}$ is standard deviation of the vibration signal and it is expressed as
(18)$$\sigma_{x}=\sqrt{\frac{1}{N-1}{\sum_{t=1}^{N}\left[x\left(t\right)-\mu \right]}^{2}},$$
where $\mu =\frac{1}{N}\overset{N}{\underset{t=1}{\sum }}\left|x\left(t\right)\right|$ is the mean of x(t).

#### 5.3.2. Mobility Parameter

The mobility parameter is defined as the ratio of the square root of variance of the first derivative of the vibration signal divided by the variance of the vibration signal or as the ratio of the standard deviation of the first derivative of the vibration signal to the standard deviation of the vibration signal.

For sampled vibration signal $x\left(t\right)=\left[x_{1},x_{2},\cdots ,x_{N}\right]$, the mobility parameter of x(t) is given by
(19)$$Mobility\left(x\left(t\right)\right)=\sqrt{\frac{Variance\left(dx\left(t\right)/dt\right)}{Variance\left(x\left(t\right)\right)}}=\frac{\sigma_{x^{\prime }}}{\sigma_{x}},$$
where $\sigma_{x^{\prime }}$ is standard deviation of the first derivative of the vibration signal.

#### 5.3.3. Complexity Parameter

The complexity parameter gives an estimate of the bandwidth of the signal and indicates how closely the vibration signal resembles a pure sine wave. It is defined as the ratio of mobility of the first derivative of vibration signal to the mobility of the vibration signal.

For sampled vibration signal $x\left(t\right)=\left[x_{1},x_{2},\cdots ,x_{N}\right]$, the complexity parameter of x(t) is given by
(20)$$Complexity\left(x\left(t\right)\right)=\frac{Mobility\left(dx\left(t\right)/dt\right)}{Mobility\left(x\left(t\right)\right)}=\sqrt{\frac{\frac{\sigma_{x^{\prime \prime }}}{\sigma_{x^{\prime }}}}{\frac{\sigma_{x^{\prime }}}{\sigma_{x}}}},$$
where $\sigma_{x^{\prime \prime }}$ is the standard deviation of the second derivative of the vibration signal.

The Hjorth parameters feature H=[Activity(x(t)),Mobility(x(t)),Complexity(x(t))] is used as part of the feature vector of rolling bearing fault identification.

In this paper, the extracted feature vectors are composed of multi-features [E, S, H] as the final feature vector.

## 6. Fault Diagnosis Model Based on MIGA-VMD and Multi-Features

On the basis of the above theory, the fault diagnosis model based on MIGA-VMD and multi-features is shown in Figure 4. The specific implementation steps are as follows:

Step 1: Taking the minimum value of the local envelope entropy as the objective function, the MIGA algorithm is used to search for the optimal parameter combination $\left[K_{0},\alpha_{0}\right]$ of VMD.

Step 2: Use VMD with parameter optimization to decompose the vibration signal in each state, and obtain $K_{0}$ IMF components respectively.

Step 3: Analyze the correlation between each IMF component and the original signal by calculating the Holder coefficient of each IMF component and the original signal.

Step 4: Calculate the Renyi entropy of the six IMF components with greater correlation with the original signal to form the Renyi entropy feature; combine the six IMF components with greater correlation with the original signal to form a matrix, and perform singular value decomposition on the matrix to form singularity Value feature; calculate the Hjorth parameters of the IMF component with the greatest correlation with the original signal to form the Hjorth parameters feature.

Step 5: The Renyi entropy feature, singular value feature and Hjorth parameters feature are composed of multi-features as the final feature vector.

Step 6: After the sample feature value is normalized to [0, 1], the feature vector of the training set is input to the classifier for training to obtain a classification model.

Step 7: Input the feature vector of the test set sample into the trained classification model to realize fault diagnosis.

## 7. Experimental Verification and Result Analysis

### 7.1. Introduction of Experiment

In order to verify the effectiveness of the proposed method in the fault diagnosis of rolling bearings, this paper uses rolling bearing data from the Electrical Engineering Laboratory of Case Western Reserve University. The rolling bearing test system selected SKF deep groove ball bearings of model 6205-2RSJEM, and the experimental bearings were damaged to varying degrees through electrical discharge machining. The vibration signal collection frequency was 12 kHz. The fault types include inner ring fault, outer ring fault and ball fault. The diameter of the fault is 7‰, 14‰, 21‰, and 28‰ respectively. The damage points of the bearing outer ring are at 3 o’clock, 6 o’clock, and 12 o’clock respectively.

When the motor load is 0 hp and the speed is 1797 r/min, the bearing vibration data for analysis is obtained. The vibration signals of 16 different types of failures, severity and damage points are analyzed. Fifty groups of vibration signals of each state are selected, including 30 groups of training samples and 20 groups of test samples. Each sample is composed of 2048 time series points. The information of bearing vibration data is shown in Figure 5.

### 7.2. Comparison and Analysis of MIGA-VMD Decomposition Effect

Taking the inner ring fault signal with a fault severity of 7‰ as an example, the multi-island genetic algorithm is used to search for the optimal parameter combination of VMD. The curve of local envelope entropy changing with the number of iterations is shown in Figure 6. Figure 6 shows that the minimum local envelope entropy of 4.1716 appeared in the eighth iteration and has been converging since then, indicating that the optimization algorithm converges fast and has strong global optimization capabilities, and is suitable for searching for the optimal parameter combination of VMD. At the minimum value of 4.1716, the corresponding parameter combination [$K_{0},\alpha_{0}$] is [10,151]. This parameter combination is introduced into the parameter settings of the VMD algorithm. Figure 7 shows the time domain waveform and spectrum of 10 IMF components after the inner ring fault signal with a fault severity of 7 ‰ is decomposed by VMD.

For the vibration signals of 16 states, the best parameter combination $\left[K_{0},\alpha_{0}\right]$ searched by the multi-island genetic algorithm is shown in Table 2.

According to the best parameter combination $\left[K_{0},\alpha_{0}\right]$ in Table 2, K and α of the VMD method are set, and the signal samples are decomposed by the VMD method of parameter optimization.

In order to highlight the ability of the MIGA-VMD method to capture fault features, the fault signal is decomposed by EMD, EEMD, and MIGA-VMD respectively, and the Teager energy operator envelope demodulation is performed on the six IMF components that have a greater correlation with the original signal. Take the inner ring fault signal with a fault severity of 7‰ as an example, as shown in Figure 8, where the inner ring fault frequency is 161.1 Hz.

In Figure 8a, only the first component’s envelope spectrum has a relatively obvious characteristic frequency peak, indicating that the remaining five IMF components are false components. In Figure 8b, only the first three components have obvious characteristic frequency peak in the envelope spectrum, indicating that the remaining three IMF components are false components. In Figure 8c, the characteristic frequency peak appears in the envelope spectrum of the six components decomposed according to the method in this paper. In addition, the envelope spectra of the first component in Figure 8a and the first two components in Figure 8b both contain a large number of noise frequencies, while the envelope spectrum of the six components in Figure 8c contains few noise frequencies. Based on this, compared with the EMD and EEMD methods, the envelope spectrum obtained by the MIGA-VMD method has fewer interference peaks and prominent main characteristic frequency. The MIGA-VMD method can effectively extract the fault features of bearing fault signals.

### 7.3. Comparison and Analysis of Feature Extraction

The Holder coefficient is used to analyze the correlation between each IMF component and the original signal. Calculate the Renyi entropy of the six IMF components that have greater correlation with the original signal to form the Renyi entropy feature. The six IMF components that have greater correlation with the original signal are formed into a matrix, and the singular value decomposition is performed on the matrix to form singular value feature. Calculate the Hjorth parameter of the IMF component with the greatest correlation with the original signal to form the Hjorth parameter feature.

Extract the Renyi entropy feature, singular value feature, and Hjorth feature from the normal signal of the rolling bearing and the fault signal of the inner ring, ball and outer ring under the fault diameter of 7‰, respectively, and use t-SNE for dimensionality reduction visualization, as shown in Figure 9a. Extract the Renyi entropy feature, singular value feature, and Hjorth parameter feature from the inner ring fault signals of different severity, and use t-SNE for dimensionality reduction visualization, as shown in Figure 9b. The Renyi entropy feature, the singular value feature, and the Hjorth parameter feature are extracted from the outer ring fault signals of different damage points, and use t-SNE for dimensionality reduction visualization, as shown in Figure 9c.

It can be seen from Figure 9a that the Renyi entropy feature and singular value feature extracted from bearing vibration signals of different fault types have better inter-class separation and intra-class aggregation than Hjorth parameter feature. It can be seen from Figure 9b that the singular value feature and Hjorth parameter feature extracted from bearing vibration signals of different severity levels have better inter-class separation and intra-class aggregation than Renyi entropy feature. It can be seen from Figure 9c that the Renyi entropy feature and Hjorth parameter feature extracted from bearing vibration signals at different damage points have better inter-class separation and intra-class aggregation than singular value feature. As Renyi entropy feature, singular value feature, and Hjorth feature show their respective advantages and disadvantages when classifying vibration signals of different fault types, different severity levels and different damage points, the Renyi entropy feature, singular value feature and Hjorth parameter feature are composed of multi-features as the final feature vector.

In the field of bearing fault diagnosis, methods such as permutation entropy, sample entropy, and energy entropy are often used to characterize the fault information of bearing signals. In order to prove the effectiveness and superiority of the multi-features feature extraction method, the MIGA-VMD method is used to decompose different bearing vibration signals. The permutation entropy feature, sample entropy feature, energy entropy feature, and multi-features are extracted from three perspectives of different fault types, different severity levels, and different damage points. The effects of the four feature extraction methods are compared through t-SNE dimensionality reduction visualization, such as shown in Figure 10.

It can be seen from Figure 10a that the energy entropy feature and multi-features extracted from the bearing vibration signals of different fault types have better inter-class separation and intra-class aggregation effects than permutation entropy feature and sample entropy feature. It can be seen from Figure 10b that the sample entropy feature and multi-features extracted from bearing vibration signals of different severity levels have better inter-class separation and intra-class aggregation effects than permutation entropy feature and energy entropy feature. It can be seen from Figure 10c that the permutation entropy feature and multi-features extracted from bearing vibration signals at different damage points have better inter-class separation and intra-class aggregation effects than sample entropy feature and energy entropy feature. In summary, multi-features have better inter-class separation and intra-class aggregation effects from three perspectives of different fault types, different severity levels, and different damage points. Compared with the permutation entropy feature, sample entropy feature, and energy entropy feature, multi-features can more accurately characterize the fault information of the bearing signal.

The MIGA-VMD method is used to decompose 16 different bearing vibration signals, multi-features are extracted and dimensionality reduction visualization is performed through t-SNE, as shown in Figure 11.

It can be seen from Figure 11 that after using MIGA-VMD decomposition and multi-features feature extraction method, the aggregation of 16 types of data features is good, and the feature distinction is obvious. Multi-features can accurately characterize the fault information of bearing signals and effectively improve the accuracy of fault recognition.

### 7.4. Comparison and Analysis of Fault Diagnosis Accuracy

In order to further verify the superiority of MIGA-VMD decomposition and multi-feature feature extraction methods, the fault classifiers in literature [23,24,25] are used to diagnose and identify 16 bearing signals. Figure 12, Figure 13 and Figure 14 and Table 3 shows the fault diagnosis results of the method proposed in this paper with three classifiers, and compared with three documents. Reference [23] proposed a fault classification method. Four kinds of bearing vibration signals were decomposed by EMD, Hjorth parameters were extracted as feature vectors, and a rule-based classifier was used to classify the vibration signals. The classification accuracy is 93.82%. Reference [24] extracts the statistical features of seven bearing vibration signals and the energy entropy features of the IMF components after EMD decomposition, and uses ANN to classify the vibration signals. The classification accuracy is 93%. Reference [25] extracts 37 features in time domain, frequency domain, and time-frequency domain from 12 kinds of bearing vibration signals, and uses PSO-SVM to classify the vibration signals with a classification accuracy of 93.33%.

In this paper, the signal is decomposed by MIGA-VMD, and the Renyi entropy feature, singular value feature, and Hjorth parameter feature are composed of multi-features as the final feature vector, which is input to the classifier for bearing fault diagnosis. A good feature extraction method can effectively extract the fault features of the bearing fault signal, which plays a vital role in improving the accuracy of fault recognition. It can be seen from Figure 12, Figure 13 and Figure 14 and Table 3 that the method proposed in this paper can not only diagnose the fault of 16 kinds of bearing signals but also has higher diagnostic accuracy than the above papers, which verifies the superiority of the method.

## 8. Conclusions

This paper proposes a feature extraction method based on MIGA-VMD decomposition and multi-features. The application of this method to the fault diagnosis of bearing signal fully verified the effectiveness of the method. Finally, the following specific conclusions can be drawn:(1)Through the envelope spectrum analysis, the MIGA-VMD method can extract the fault feature information of the signal more effectively and significantly than EMD and EEMD, which is of great significance for the fault diagnosis of rolling bearings.(2)Through t-SNE dimensionality reduction visualization analysis, compared with the permutation entropy feature, sample entropy feature, and energy entropy feature, the multi-features composed of Renyi entropy feature, singular value feature, and Hjorth parameter feature can more accurately characterize the fault information of the bearing signal and effectively improve the accuracy of fault recognition.(3)The MIGA-VMD decomposition and multi-features feature extraction method is combined with three classifiers respectively to establish new fault diagnosis models. Through experimental comparison and analysis, the new fault diagnosis models have high recognition accuracy, which verifies the superiority of the method proposed in this paper.

## Figures and Tables

**Figure 1 entropy-22-00995-f001:**
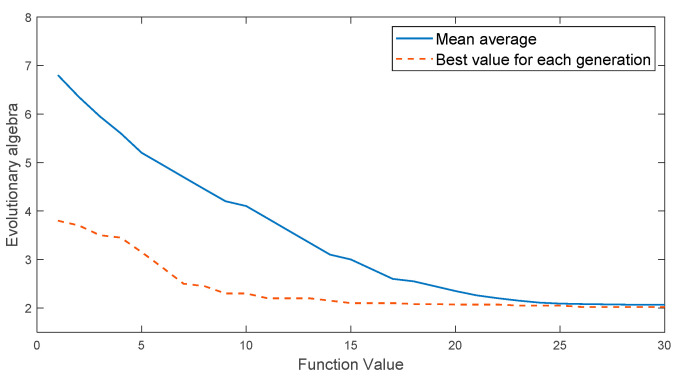
Iterative curve of the optimization process of genetic algorithm.

**Figure 2 entropy-22-00995-f002:**
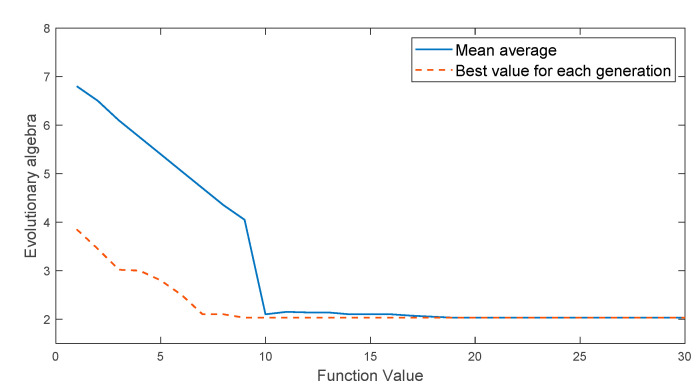
Iterative curve of the optimization process of multi-island genetic algorithm.

**Figure 3 entropy-22-00995-f003:**
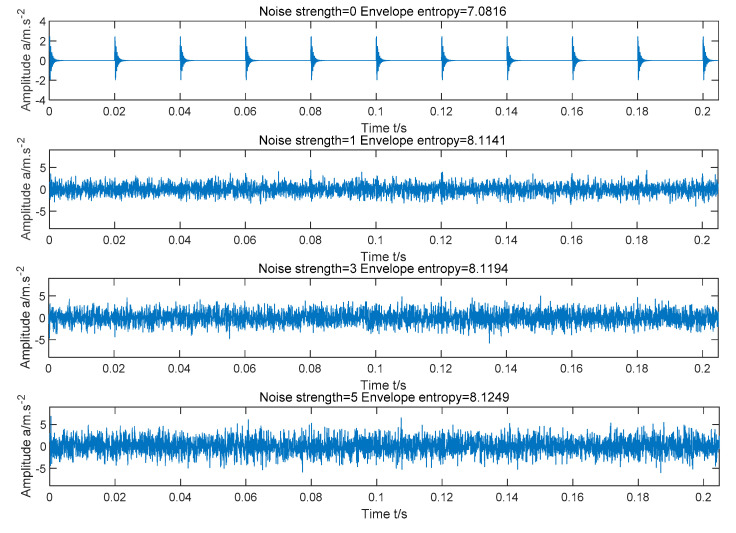
Simulation signal time domain waveform under different noise strength.

**Figure 4 entropy-22-00995-f004:**
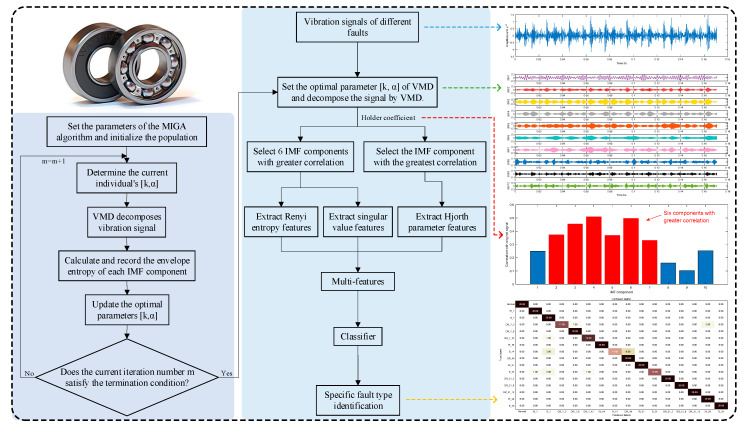
Flow chart of the proposed method.

**Figure 5 entropy-22-00995-f005:**
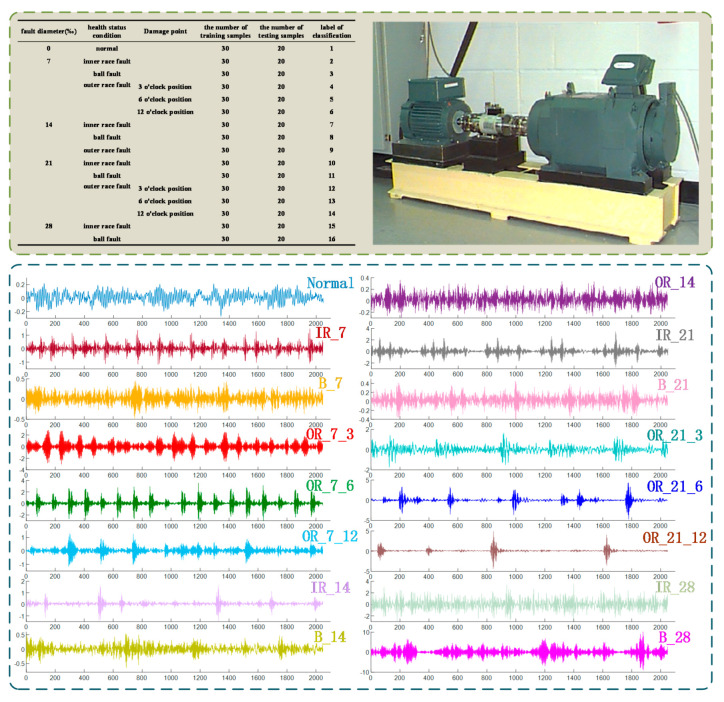
Information of bearing vibration data.

**Figure 6 entropy-22-00995-f006:**
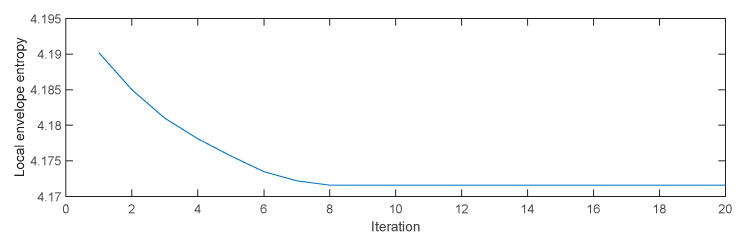
Iterative curve of the optimization process.

**Figure 7 entropy-22-00995-f007:**
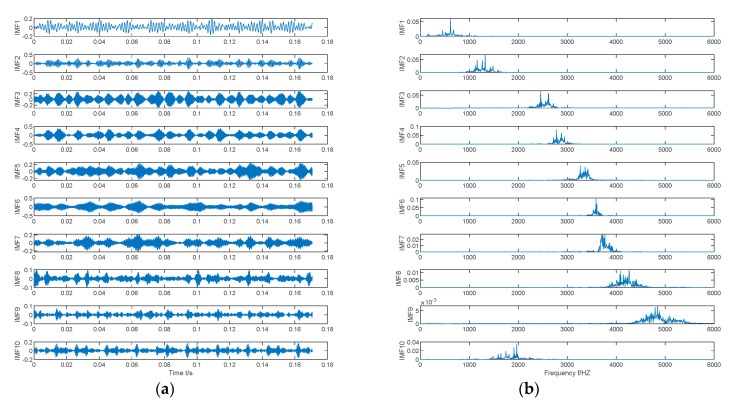
Waveform and spectrum of 10 IMF components. (**a**) Time domain, (**b**) FFT spectrum.

**Figure 8 entropy-22-00995-f008:**
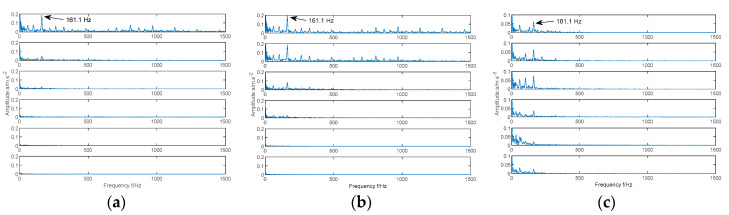
Teager energy operator envelope spectrum of six IMF components. (**a**) EMD, (**b**) EEMD, (**c**) MIGA-VMD.

**Figure 9 entropy-22-00995-f009:**
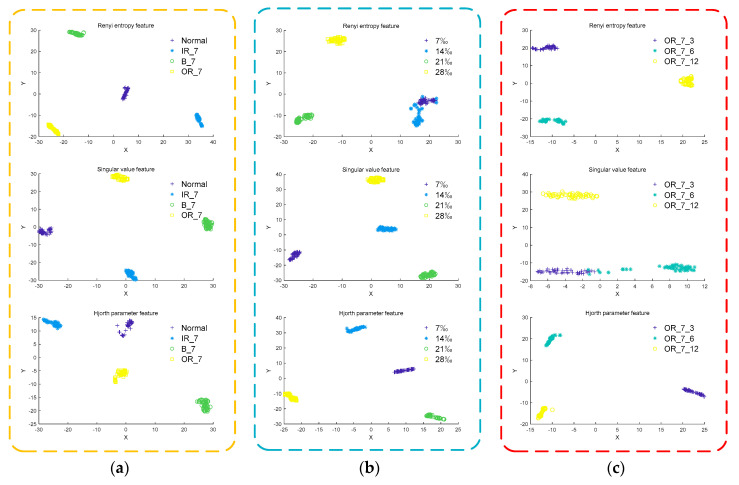
Low-dimensional fault feature distribution. (**a**) Different fault types; (**b**) different severity; (**c**) different damage points.

**Figure 10 entropy-22-00995-f010:**
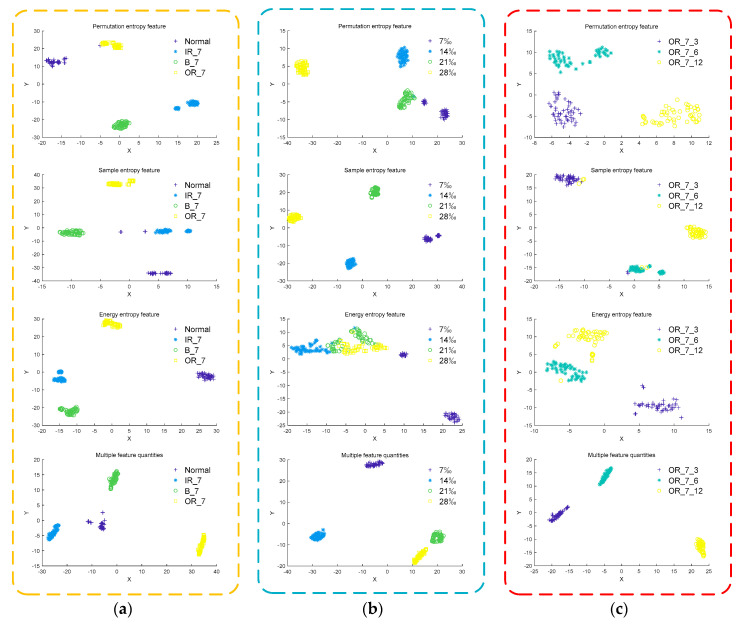
Low-dimensional fault feature distribution. (**a**) Different fault types; (**b**) different severity; (**c**) different damage points.

**Figure 11 entropy-22-00995-f011:**
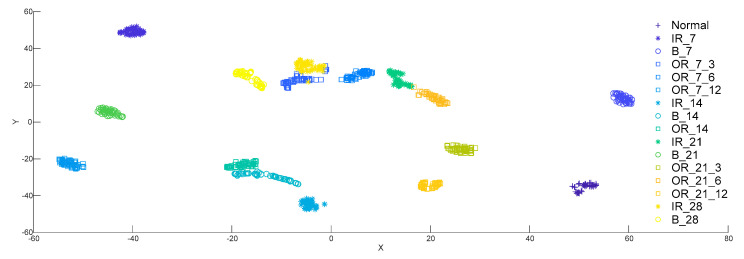
Low-dimensional feature distribution of 16 kinds of bearing signals.

**Figure 12 entropy-22-00995-f012:**
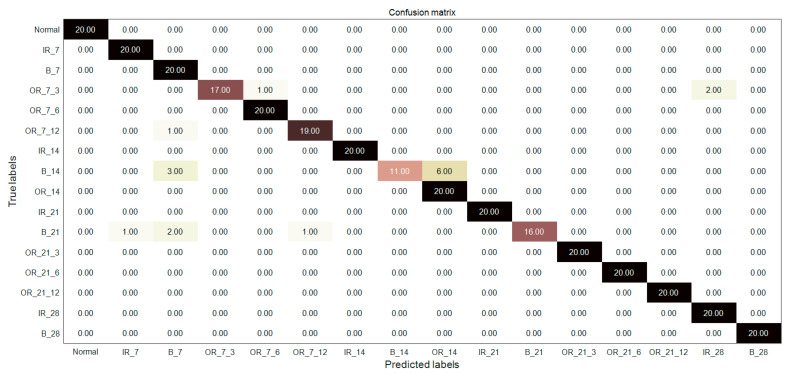
Diagnosis results of the proposed feature extraction method combined with the classifier in [23].

**Figure 13 entropy-22-00995-f013:**
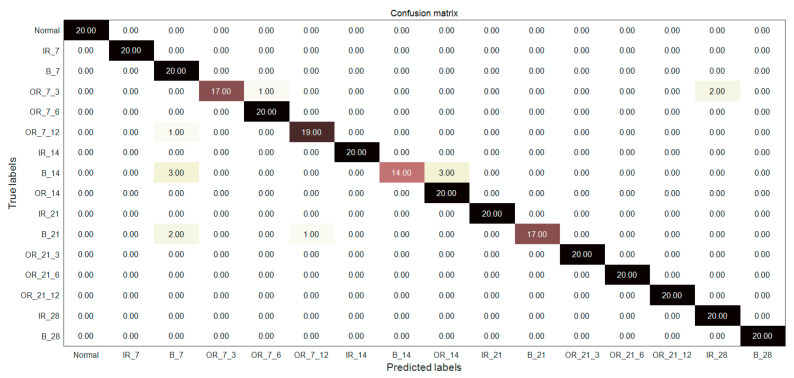
Diagnosis results of the proposed feature extraction method combined with the classifier in [24].

**Figure 14 entropy-22-00995-f014:**
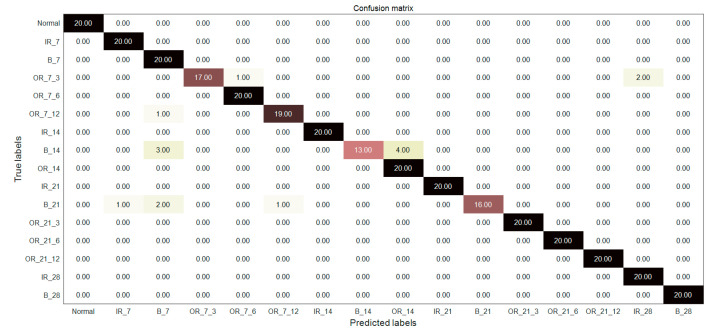
Diagnosis results of the proposed feature extraction method combined with the classifier in [25].

**Table 1 entropy-22-00995-t001:** Algorithm table for multi-island genetic algorithm.

Algorithm: multi-island genetic algorithm
**Step1:** Initialize population $P_{0}$. **Step2:** The initial population $P_{0}$ is divided into multiple “islands”, namely subpopulations. **Step3:** Calculate the fitness value of each individual on each “island”. **Step4:** The selection, crossover, and mutation operations in the standard genetic algorithm are performed on each island. **Step5:** If the migration conditions are met, then migrate from one island to another. **Step6:** If the current number of iterations n is less than the maximum number of iterations MaxGen, then proceed to Step3, otherwise proceed to Step7. **Step7:** The individual with the best fitness value in the output population is the optimal solution.

**Table 2 entropy-22-00995-t002:** Optimal parameter combination $\left[K_{0},\alpha_{0}\right]$.

Health Status Condition	$\left[K_{0},\alpha_{0}\right]$	Health Status Condition	$\left[K_{0},\alpha_{0}\right]$
Normal	[12,94]	OR_14	[14,510]
IR_7	[10,151]	IR_21	[15,2519]
B_7	[14,248]	B_21	[14,1246]
OR_7_3	[13,935]	OR_21_3	[13,3987]
OR_7_6	[15,3876]	OR_21_6	[14,3419]
OR_7_12	[14,921]	OR_21_12	[13,1065]
IR_14	[9,3241]	IR_28	[15,3968]
B_14	[14,576]	B_28	[14,678]

**Table 3 entropy-22-00995-t003:** Comparison of fault diagnosis results between the method proposed in this article and the methods of three documents.

Literature	Feature	Classifier	No. of Classes	Accuracy
[23]	EMD Hjorth parameter features	Rule-based classifiers	4 (N, IR, B, OR)	93.82%
This Work	MIGA-VMD-multi-features	Rule-based classifiers	16	94.68%
[24]	EMD energy entropy of the first seven IMFs and classical statistical features	ANN	7(H, DR, FR, DIR, FIR, DOR, FOR)	93%
This Work	MIGA-VMD-multi-features	ANN	16	96.25%
[25]	16 Time domain +13 Frequency domain + 8 Time-frequency domain features	PSO-SVM	12(N, IR7, IR14, IR21, IR28, OR7, OR14, OR21, B7, B14, B21, B28)	93.33% with time-domain features
This Work	MIGA-VMD-multi-features	PSO-SVM	16	95.31%

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
