# Peer review of "A Novel Method Based on Multi-Island Genetic Algorithm Improved Variational Mode Decomposition and Multi-Features for Fault Diagnosis of Rolling Bearing"

_entropy, 2020, doi:10.3390/e22090995_

Round 1

Reviewer 1 Report

The paper presents relevant results, and contributes to research on signal analysis, and damage detection in rotating systems. The authors propose a fault diagnosis method, obtained from the VMD method, the method being improved using optimization through genetic algorithm (GA).
The paper is able to be published after minor corrections. It is necessary to justify in the paper why GA was used, and not another optimization technique. (For example, why didn't they use PSO?).

Author Response

Thank you for your review, please see the attachment for the revised content.

Reviewer 2 Report

Dear Authors,

I have some comments on your article:

  1. At the end of the Introductions section, information on how the article is organized is missing.
  2. Please check all equations, symbols, and index used in the text and equations.
  3. The font size in equations is too large. Please edit all equations.
  4. Figure 2. The flow chart of the proposed method - The drawings in the third column are illegible.
  5. Similarly, the tables in Figures 10, 11, and 12.
  6. Literature should be checked if there are no newer items. Especially from the last 18 months. It would be good to add several references.
  7. How the authors assess the possibility of increasing the accuracy of the assessment of damage to rolling bearings. Maybe it's worth writing it down in the summary.

Author Response

(The authors gave the same response as above.)

Round 2

Reviewer 2 Report

Dear Authors,

Thank you very much for introducing changes that have improved the quality of the article. I have no more comments.

Best regards,
MS